# From Emotional Abuse to a Fear of Intimacy: A Preliminary Study of the Mediating Role of Attachment Styles and Rejection Sensitivity

**DOI:** 10.3390/ijerph21121679

**Published:** 2024-12-17

**Authors:** Ricky Finzi-Dottan, Hila Abadi

**Affiliations:** School of Social Work, Bar Ilan University, Ramat Gan 5290002, Israel; abadihila@gmail.com

**Keywords:** emotional abuse, attachment styles, rejection sensitivity, fear of intimacy, mediation

## Abstract

Based on the literature indicating that emotional abuse erodes children’s secure attachment bonds, this study aimed to examine a mediation model positing that insecure attachment (anxious and/or avoidant styles) would mediate the association between childhood emotional abuse and rejection sensitivity (rejection and acceptance expectancies), which, in turn, will be associated with a fear of intimacy. One hundred and eighty members of the Israeli public aged 21–30 who were in relationships participated in the study. The results showed that avoidant and anxious attachment mediated the relationship between childhood emotional abuse and a fear of intimacy, acceptance expectancy mediated the relationship between avoidant and anxious attachment and a fear of intimacy, anxious attachment mediated the relationship between childhood emotional abuse and both rejection and acceptance expectancies, and avoidant attachment mediated the relationship between childhood emotional abuse and acceptance expectancy. These results indicated the salient role of insecure attachment styles and their associations with rejection sensitivity in forming intimate relationships due to having experienced childhood emotional abuse. Intervention promoting “earned security” was recommended.

## 1. Introduction

Intimacy is highly valued in interpersonal relationships. Partners are more likely to feel intimacy with each other when they can both discuss their vulnerabilities, mutually validating and supporting each other’s self-disclosure. This process may be challenging to some individuals who find it difficult to share their inner worlds [1]. Various developmental theories, including attachment theory [2], predict how poor parental care, or maltreatment in infancy and childhood, constitute a major risk factor in psychosocial development, affecting both the ability to relate and form intimate relations and one’s psychological well-being in adulthood [3]. Childhood abuse has long been recognized as a risk factor for good intimate relationships [4]. It comes in many forms, one of which is emotional abuse. The latter often accompanies physical or sexual abuse but also frequently occurs in isolation, making it the most pervasive and chronic type of maltreatment. While its long-term consequences have been increasingly acknowledged [5], this type of abuse and its effect on adult life has been the subject of fewer studies compared to physical or sexual abuse. Riggs (2010) [5] was among the first to examine the link between emotional abuse, attachment styles, and adult interpersonal relations. Other studies have linked emotional abuse to difficulties in interpersonal and romantic relationships [5,6,7]. However, these findings have not been articulated as an integrative framework, nor has the intervening variable in this process been studied. The present study, therefore, seeks to fill this gap by exploring the association between childhood emotional abuse and insecure attachment styles affecting rejection sensitivity, which, in turn, would be linked to a fear of intimacy. It further seeks to examine how insecure attachment and rejection sensitivity mediate this link between emotional abuse and a fear of intimacy.

Emotional abuse consists of recurrent parental critical attacks, rejection, devaluation, contempt, and ignoring the child, all of which undermine the child’s emerging identity [6]. At a young age, it also includes emotional neglect, i.e., caregivers’ irresponsibility or failure to satisfy their children’s basic psychological needs for love, belongingness, nurturance, and support [7]. Individuals who experience emotionally abusive relationships in childhood are at a distinct disadvantage in interpersonal contexts because they develop a distorted understanding of what loving and caring relationships involve [8,9,10].

Attachment theory offers insights into the mechanisms that link childhood maltreatment to later outcomes [11]. It posits that early interactions with one’s primary caregiver shape one’s internal working models of relationships, which serve as the prism for relational patterns throughout life. Repeated interactions with available, sensitive, and responsive attachment figures create a persisting sense of attachment security, as well as positive working models of self (as being lovable) and others (as being reliable, helpful, and loving). However, when caregivers are frequently rejecting, belittling, or unresponsive in times of need, attachment security is undermined, and serious doubts arise about others’ availability and trustability, as well as about one’s own value and lovability, resulting in insecure attachment (e.g., [12]). According to attachment theory, individual differences in attachment histories result in a specific attachment orientation, which tends to be organized around two main attachment dimensions: anxiety and avoidance. Attachment anxiety is characterized by a fear of abandonment and a constant need for reassurance of others’ reliability and availability in times of need. Attachment avoidance is marked by distrust, emotional distance, and a reluctance to rely on others [12]. Individuals scoring low on both dimensions are securely attached and tend to have adaptive strategies for modulating emotions, trust, and the ability to form satisfactory intimate relationships [13]. In contrast, individuals who experience emotionally abusive attachment relationships in childhood, such as punitive, rejecting parenting behaviors, may be more likely to expect rejection and be biased to perceived slights in subsequent interactions with others later in life [14].

Euteneuer and colleagues (2024) [15] suggest that forms of emotional maltreatment (emotional abuse and neglect) are especially linked to rejection sensitivity in emerging adults. Rejection sensitivity is a potential cognitive–affective perceptual bias referring to a lack of feeling of acceptance by others, as well as a disposition to anxiously expect, readily perceive, and intensely react to rejection by significant others. Individuals reporting high acceptance expectancy tend to expect that others would be helpful and supportive in times of need. However, situations that require one to express their need for support to a significant other could activate general anxiety over whether or not the latter would indeed meet one’s need for acceptance or reject them. Individuals high in rejection expectancy tend to be hypervigilant for signs of rejection. When they encounter rejection cues, however minimal or ambiguous, they readily perceive intentional rejection, thereby experiencing feelings of rejection [16,17]. Downey and Feldman (1996) [16], based on attachment theory [2], have proposed that when parents tend to meet children’s expressed needs with hostility or rejection, their children become sensitive to rejection. Thus, they develop the expectation that they will probably be rejected when seeking acceptance and support from significant others and learn to place a particularly high value on avoiding such rejection. Furthermore, they experience anticipatory anxiety when expressing needs or vulnerabilities to significant others, perceiving intentional rejection in the ambiguous behavior of others.

Whereas rejection sensitivity may originally develop as a self-protective reaction to parental rejection, this system may prompt behaviors that are poorly adapted to adult circumstances [18]. When activated in a relatively benign social world, rejection sensitivity may lead people to behave in ways that undermine their chances of maintaining a supportive and satisfying close relationship [19]. Rejection expectancies may establish a self-fulfilling prophecy wherein individuals would be led to behave in ways that would elicit rejection from their partners [20]. A fear of rejection was found to be negatively associated with the intimacy-building process and especially predictive of a fear of intimacy [21].

Fear of intimacy is “the inhibited capacity of an individual, because of anxiety, to exchange thoughts and feelings of personal significance with another individual who is highly valued” [22] (p. 219) and tends to oppose the ability to form close and intimate relationships [23,24]. Greenberg and Goldman (2008) [25] view a fear of intimacy as a self-protective stance driven by the vulnerability of needing others, getting rejected, hurt, or being embarrassed. For children who experience a rejecting, cold, or uninvolved caregiver, a fear of intimacy can be an adaptive behavior [26].

To summarize, we assume that experiences of childhood emotional abuse resulting in insecure attachment and viewing others as untrusted [5] would be associated with protection effects from possible exposure to rejection [27], consequently leading to a fear of close, intimate relations [28].

### 1.1. The Present Study

The present study seeks to explore a mediating model for expanding the knowledge of the mechanisms that link childhood emotional abuse to later adult interpersonal relations. The literature reviewed suggests a chain of associations between emotional abuse in childhood and impairment in forming a sense of security and secure attachment bonds, resulting in insecure attachment styles (whether avoidant or anxious). Emotional abuse and insecure attachment styles instill distrust in the goodwill of others, as well as in the availability of their support in times of need, resulting in rejection sensitivity and anxiety about intimate relations, namely, a fear of intimacy.

Several studies have examined these variables—childhood emotional abuse and attachment styles, as well as their association with rejection sensitivity and fear of intimacy—separately; however, the contribution of our study lies in its examination of the parallel and serial mediation of all these variables.

### 1.2. We Hypothesize That

**Hypothesis 1:** 
*Insecure attachment (anxious and/or avoidant) mediates the association between childhood emotional abuse and rejection sensitivity (rejection and/or acceptance expectancies).*


**Hypothesis 2:** 
*Insecure attachment (anxious and/or avoidant) mediates the association between childhood emotional abuse and rejection sensitivity (rejection and or acceptance expectancies), which, in turn, is associated with fear of intimacy.*


## 2. Materials and Methods

### 2.1. Participants

The study participants included 126 women and 54 men (n = 180) from the general Israeli population. The average age of all participants was 28.18 (*sd* = 3.39); the men’s average age was 27.95 (*sd* = 3.60), and the women’s was 28.20 (*sd* = 3.46). All participants had been in a relationship for three or more years. A total of 21% had children, compared to 79% who were childless. As for education, 18.30% had a high school education, and most of the participants—81.10%—had an academic education. In terms of religious orientation, about 67.2% defined themselves as secular, and 32.8% as traditional/religious. As for economic status, 5.0% defined their economic situation as poor, 47.2% as moderate, 38.9% as good, and 8.9% as very good.

### 2.2. Questionnaires

Childhood emotional abuse was assessed using the Childhood Trauma Questionnaire (CTQ-28), which has demonstrated good criterion-related validity [7]. This 28-item questionnaire uses a 5-point Likert scale ranging from 1 (never true) to 5 (very often true) and captures five forms of childhood abuse: emotional abuse, physical abuse, sexual abuse, emotional neglect, and physical neglect. The reliability and validity of the Hebrew version of the CTQ have been demonstrated (Finzi-Dottan and Karu (2006) [6]). Similarly to Finzi-Dottan and Karu’s (2006) [6] study and that of Finzi-Dottan and Harel (2014) [29], the present study also found high correlations (*r* = 0.62) between emotional neglect and emotional abuse, and therefore, these two measures were consolidated under a single dimension, referred to as emotional abuse.

Given our particular interest in the effects of emotional maltreatment, we focused on the emotional abuse and emotional neglect subscales. Sample items from these subscales include “People in my family called me things like ‘stupid’, ‘lazy’, or ‘ugly’” and “I felt loved” (reverse coded). We added up the scores on these subscales to obtain a measure of emotional abuse. Goodman and colleagues (2014) [30] also used a combination of scores from the emotional abuse and neglect subscales of the CTQ to quantify emotional maltreatment. In the present study, 52 of the participants (29%) reported having undergone emotional abuse, and a high internal consistency was found for this combined measure (Cronbach’s α = 0.93).

Attachment styles were assessed using the Experience in Close Relationships (ECR) scale—a 36-item measure designed to assess adult attachment styles that was developed by Brennan and colleagues (1998) [31]. Respondents indicate the degree to which a statement describes their own experience in close relationships on a 7-point scale ranging from 1 (not at all) to 7 (very much). The scale yields scores on two attachment dimensions: one reflecting anxious attachment orientation (18 items, e.g., “I need a lot of reassurance that close relationship partners really care about me”) and avoidant attachment orientation (18 items, e.g., “I do not feel comfortable opening up to other people”). The two measures generated from this questionnaire were calculated by generating an average from participants’ scores on the 18 items of each index to reflect avoidant and anxious attachment. Low scores on both dimensions represent a more secure attachment style. The ECR was translated into Hebrew by Mikulincer and Florian (2000) [32], who also validated its two-factor structure among an Israeli sample. In this study, the internal consistencies of the two were Cronbach’s α = 0.82 for avoidant attachment and Cronbach’s α = 0.80 for anxious attachment.

Rejection Sensitivity was assessed using the Rejection Sensitivity Questionnaire (RSQ) [16], which evaluates general expectations and anxiety about whether significant others will meet one’s need for acceptance or reject it. The questionnaire presents respondents with 18 situations in which they make a request of another person (i.e., “You are asking someone at your place of work or studies to help you cope with an urgent task”). For each scenario, participants are asked to use a 6-point Likert scale to rate two items: (i) whether they would be concerned or anxious about the response to their request (rejection expectancy), and (ii) whether they would expect the other person to honor or reject the request (acceptance expectancy). Taubman-Ben-Ari and colleagues (2002) [33] adapted the RSQ to the young Israeli adult population and demonstrated the reliability and validity of the Hebrew version.

The two measures generated from this questionnaire were calculated using an average of the scores yielded by the 18 items of each index to reflect rejection and acceptance expectancies. The internal consistencies of the two scales were Cronbach’s α = 0.89 for rejection expectancy and Cronbach’s α = 0.89 for acceptance expectancy.

Fear of intimacy was assessed using the Fear of Intimacy Scale (FIS) [22]. This 35-item questionnaire measures the degree to which participants are uncomfortable with or fear intimacy in their relationships. Participants are asked to think of their current intimate relationship when responding to each item. Example items include “I would feel comfortable expressing my true feelings to [the other person]” and “I might be afraid to reveal my innermost feelings to [the other person]”. Participants use a 5-point Likert scale from 1 (not at all true of me) to 5 (extremely true of me) to respond to each item. Taubman-Ben-Ari (2004) [34] adapted the RSQ to the young Israeli adult population and demonstrated the reliability and validity of the Hebrew version.

The score on the questionnaire was calculated by adding up the scores of the items so that a high score meant a greater fear of intimacy. In this study, Cronbach’s alpha was α = 0.93.

### 2.3. Procedure

The study was approved by the School of Social Work’s Ethics Committee at the authors’ university (protocol code #122106). An online survey was conducted among a convenience sample, for which participants were recruited in two ways: by advertising on various social network platforms (e.g., Facebook pages and social network groups) and by using snowball sampling techniques such as sending the study link to a list of e-mail contacts and then asking each contact to forward the e-mail to his or her contacts (such as students or other interest groups). All questionnaires used in the present study were administered in Hebrew. Data were collected online using Qualtrics, a secure web-based survey data collection software. It took 20 min to complete, on average, and was open from December 2022 to December 2023. The survey was anonymous, and no data linking participants to recruitment sources were collected. When potential respondents clicked on the link to the survey, they were guided to a page that provided information about the purpose of the study and nature of the questions, as well as a consent form. In the present study, no incentives were offered. Since the questionnaires were filled out using Qualtrics, a platform in which questionnaires that are not fully answered are not saved or submitted, no data were missing.

### 2.4. Data Analysis

The data were analyzed using SPSS version 29 [35] and divided into several phases. First, the averages and standard deviations of the study variables were examined, followed by the production of Pearson correlation coefficients corresponding to the variables to explore the initial associations between them. Next, a hierarchical regression analysis was conducted using the PROCESS package [36] in conjunction with SPSS version 29 [35] to predict fear of intimacy. Finally, a PROCESS analysis was also conducted [36] to test the possibility that attachment styles and rejection sensitivity mediate between reports of childhood emotional abuse and fear of intimacy.

## 3. Results

To examine the pattern of associations between the main study measures, we conducted a series of Pearson correlations. Means and standard deviations are presented in Table 1, followed by correlation coefficients. The analyses indicated that the higher the reports of childhood emotional abuse, anxious and avoidant attachment, and rejection expectancy were, the higher the fear of intimacy, and the lower the acceptance expectancy was, the higher the fear of intimacy. Finally, relationship duration was found to have no effect on fear of intimacy.

To test the relationships between the predicting variables and the fear of intimacy, including potential indirect effects and mediation, we constructed a four-step regression model (see Table 2) using process package 4.3 [32]. Overall, the model explained 41.7% of the variance. As part of the first step, the participants’ gender was introduced. The analysis indicated that women and men do not differ in fear of intimacy; however, women did report insignificantly higher scores. In the second step, childhood emotional abuse was introduced, adding 17.6% to the prediction. In step 3, attachment styles were added to the model, emerging as stronger in magnitude for avoidant attachment. A decline in the contribution of emotional abuse was also noted, suggesting that avoidant and anxious attachment are significant mediators. Finally, rejection and acceptance expectancy were introduced into the regression model, revealing that only acceptance expectancy contributed to the prediction of a fear of intimacy. In this step, the contribution of emotional abuse continued to decrease, as did the contribution of avoidant attachment slightly, while anxious attachment became insignificant, suggesting that they play a mediating role.

In order to examine the mediating hypotheses, the PROCESS function version [36] was used, which enables the estimation of coefficients in a mediation model using a bootstrapping method for estimating confidence intervals. The study employed 5000 bootstrap samples to calculate bias-corrected 95% confidence intervals. Table 3 displays the coefficients and their confidence intervals for the significant results found. Figure 1 shows that seven indirect effects were found.

Emotional abuse indirectly affects the fear of intimacy through its effect on avoidant attachment (indirect β = 0.32 [CI = 0.15, 0.50]) and anxious attachment (indirect β = 0.05 [CI = 0.00, 0.13]); emotional abuse is positively related to avoidant or anxious attachment, which positively affects fear of intimacy. Avoidant and anxious attachment indirectly affects the fear of intimacy through its effect on acceptance expectancy (respectively, indirect β = 0.08 [CI = 0.03, 0.15]; indirect β = 0.05 [CI = 0.01, 0.11]); avoidant and anxious attachment are negatively related to acceptance expectancy, which negatively affects the fear of intimacy. Furthermore, three indirect effects between emotional abuse and acceptance and rejection expectancies were found: emotional abuse indirectly affects acceptance expectancy through its effect on anxious attachment (indirect β = −0.06 [CI = −0.14, −0.00], and avoidant attachment (indirect β = −0.20 [CI = −0.34, −0.07]); emotional abuse is positively related to avoidant or anxious attachment, which negatively affects acceptance expectancy. Emotional abuse indirectly affects rejection expectancy (indirect β = 0.10 [CI = 0.005, 0.20] through its effect on anxious attachment; emotional abuse is positively related to anxious attachment, which positively affects rejection expectancy.

## 4. Discussion

The purpose of the current study was to take a deeper look at the mechanism linking childhood emotional abuse and later outcomes in adulthood in terms of forming intimate relations. Based on the literature portraying the detrimental effects of childhood emotional abuse on interpersonal relationships, we hypothesized that recollections of childhood emotional abuse would be associated with insecure attachment (avoidant or anxious). We further postulated that these, in turn, would be linked to rejection sensitivity and affect the fear of intimacy. The results supported most of these postulations. Emotional abuse was indeed found to be positively related to avoidant or anxious attachment, which had a positive effect on fear of intimacy. Furthermore, avoidant or anxious attachment was negatively related to acceptance expectancy, which negatively affected the fear of intimacy. Emotional abuse was found to be positively related to avoidant or anxious attachment, which had a negative effect on acceptance expectancy. Finally, emotional abuse was positively related to anxious attachment, which positively affected rejection expectancy.

In several studies, Riggs (2010) [5] demonstrated a clear relationship between childhood emotional abuse and the development of an insecure attachment style (whether anxious or avoidant), which, in turn, affects the quality of interpersonal relationships. Childhood emotional abuse as a parenting pattern can convey to the child that he or she is worthless, unloved, flawed, or unwanted, leading to pernicious, destructive consequences. Importantly, emotional abuse is a process, a continuous pattern of relationships with parents that instills damaging beliefs about the self (e.g., “I am stupid” or “I am not worthy of attention”). These, in turn, may result in maladaptive models of self, of others (“nobody cares for me, and no one is trustable”), and self-in-relation to others [34]. These beliefs about self and others and the resulting sense of vulnerability can place the attachment system in a continually activated state, keeping a person’s mind preoccupied with threats and the need for self-protection [9]. As a result, the attachment system has to be adjusted, leading to the adoption of hyperactivation and deactivation strategies, namely, anxious or avoidant attachment. Both anxious and avoidant styles engrain suspiciousness about the goodwill of others [12] and, as a form of self-protection reaction, may instill rejection sensitivity. The latter then renders close interpersonal interactions challenging because of the looming risk of suffering rejection [18].

The results indicated that both anxious and avoidant attachment mediate emotional abuse and acceptance expectancy, with avoidant attachment emerging stronger in magnitude. Individuals with anxious attachment employ hyperactivation strategies, hoping for acceptance and fearing being rejected at the same time. They “move”, as Mikulincer and colleagues (2010) [13] have described, in the push and pull movements of ambivalence, seeking to be loved and accepted. As mentioned, individuals with avoidant attachment employ deactivation strategies, distrust others, and view their relationships with others as non-benevolent [12]. Therefore, they probably do not expect acceptance from others. In this vein, the results revealed that acceptance expectancy mediates the relationship between avoidant and anxious attachment and the fear of intimacy. The higher the avoidant or anxious attachment is, the lower the acceptance expectancy, which affects the fear of intimacy. Despite the different strategies of avoidant and anxious attachment, they were both associated with low expectancies for acceptance. Individuals with both avoidant and anxious attachments had a negatively distorted perception of their relationships and negatively interpreted the actions of those around them. This triggers an overreaction of the defense motivation system, inducing emotions and behaviors for self-protection. The subsequent result is a fear of being rejected and not having their needs accepted [18,19]. Individuals with avoidant attachment learn to be self-reliant as a self-protective reaction to parental rejection or contempt and therefore do not expect acceptance. Furthermore, they tend to avoid close relationships and seem reluctant to be involved in close intimate relations [12]. Individuals with anxious attachment overestimate the threat of being rejected or abandoned. They desire acceptance and fear being rejected [13], leading to a fear of intimacy [28].

Finally, both avoidant and anxious attachment mediated the relationship between childhood emotional abuse and the fear of intimacy; however, avoidant attachment had a stronger magnitude. The latter serves as a defensive strategy, whereby the attachment system is deactivated by denying attachment needs, distancing emotion, and making great efforts to achieve self-reliance [2]. Children who experience emotional abuse learn to deny their attachment needs and emotions. They downplay any threats of being hurt again and avoid hostile reactions from the caregiver [5]. In adulthood, they protect themselves by deactivating attachment needs. They avoid intimacy and dependence in close relationships while maximizing emotional distance from others. Intimacy poses a threat to such individuals’ sense of control. It requires emotional closeness and self-disclosure, which they have learned can pose the risk of hurting them. Thus, they fear close intimate contact, placing great emphasis on the need to limit closeness and intimacy [6], and consequently display fear of intimacy [23,24,37]. Moreover, avoidant individuals interpret their partners’ support as unhelpful and noncaring. As a rejection-regulative mechanism, they avoid intimate relationships altogether [38]. Conversely, those with anxious attachment are highly worried about being abandoned or unloved, leading them to be ambivalent in their relational tendencies. Their desire to be close to their relationship partners while also fearing rejection [13] (Mikulincer et al., 2010) results in the fear of intimacy [26]. Similarly, research indicates that individuals with anxious attachment exhibit less closeness and trust in relationships associated with rejection sensitivity and, in turn, with a fear of intimacy [39].

### 4.1. Limitations

Although this study has important implications for understanding the processes underlying emotional abuse and interpersonal relationships, its results should be viewed with caution. Due to its cross-sectional design, the present study precludes conclusions about the direction of causality. In addition, its sampling method was convenient rather than representative, which means, for example, that respondents were typically highly educated and primarily of high socioeconomic status. The results may not, therefore, be generalized to apply to populations that are less educated and of a lower SES.

Moreover, the use of a convenient sample, non-random recruitment, impairs the generalization of the study. Thus, this study can be viewed as a preliminary one. In future studies, more robust sampling procedures should be used, ideally random sampling, to strengthen this model and the subsequent conclusions. In addition, similarly to other studies, e.g., [9,11], 70% of the participants were women in our study. Another limitation pertains to the instruments used in this study: all were self-report questionnaires, inherently subject to response bias. Moreover, the use of digital platforms—participants were recruited via social networks and reported their recollections of childhood abuse in CTQ—could also bias the results of this study. Finally, the self-reports of childhood emotional abuse recollections could lead to memory bias when filling out the questionnaire since these memories may be affected or change throughout an individual’s life.

### 4.2. Conclusions and Clinical Implications

Emotional abuse may be the most prevalent form of child maltreatment, but it is also the most hidden, elusive, and under-reported. Moreover, its consequences are severe and last throughout one’s life. In our study, 29% of the participants reported experiencing emotional abuse. This rate is similar to another study conducted in Israel (21.1%) [6], in which the indices of abuse and emotional neglect were consolidated. Aslam and colleagues’ (2024) [40] study revealed that between one-quarter and one-third of adults in the United States have experienced childhood emotional abuse (the range is 20.7–32% in most states). Stoltenborgh and colleagues’ (2012) [41] meta-analysis reported that the combined prevalence of emotional abuse for the total set of studies was 26.7%. Presumably, due to under-reporting and difficulty in definitions, this figure is the metaphorical tip of the iceberg.

Our results suggest that childhood emotional abuse has a profound impact on the attachment system, which affects individuals’ perceptions of whether their needs would be supported and accepted later in life or rejected. This vulnerability may result in a fear of intimacy. The results warrant interventions targeted at the negative perceptions of the self and others in interactions. Psychotherapy may help to ameliorate such distorted beliefs and could provide an effective context for developing more positive models of self and others. A positive change in these perceptions of the self and others is more commonly known as “earning security”. A beneficial therapeutic relationship will endorse intrapsychic changes to the identity and sense of worth, assist in making peace with the past, forgiving oneself for his/her helplessness and for blaming oneself for the abuse while promoting a sense of security and growth beyond the injuries one had experienced at the hands of their parents. Moreover, such a therapeutic relationship may lead to forming trustful intimate relationships [42].

## Figures and Tables

**Figure 1 ijerph-21-01679-f001:**
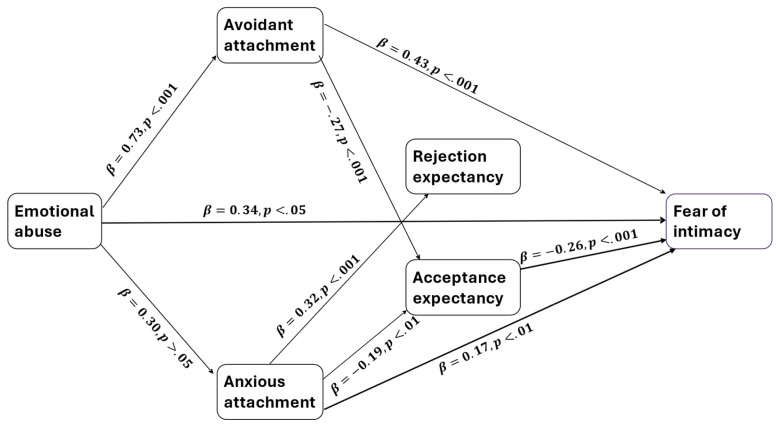
Results of the mediation model.

**Table 1 ijerph-21-01679-t001:** Descriptive statistics and correlations between study variables.

	M	SD	1	2	3	4	5	6	7
1. Emotional abuse	18.65	7.00	1						
2. Avoidant attachment	3.30	1.0	0.31 **	1					
3. Anxious attachment	3.37	1.1	0.25 **	0.15 *	1				
4. Rejection expectancy	2.83	0.96	0.32 **	0.19 **	0.34 **	1			
5. Acceptance expectancy	4.74	0.80	−0.45 **	−0.39 **	−0.23 **	−0.32 **	1		
6. Fear of intimacy	69.38	22.45	0.43 **	0.54 **	0.26 **	0.28 **	−0.48 **	1	
7. Gender	*	--	0.07	0.08	0.03	−0.15 *	−0.16 *	0.13	1
8. Relationship duration	4.51	0.62	0.01	0.03	−0.03	−0.07	−0.07	−0.12	−0.05

Note: * *p* < 0.05, ** *p* < 0.01; men—n = 54; women—n = 126. Men—1 = +; women—0 = −.

**Table 2 ijerph-21-01679-t002:** Regression coefficients for predicting fear of intimacy as a function of emotional abuse, attachment styles, and rejection sensitivity.

		b	SE	β	t
Step I	Gender	6.98	4.15	0.13	1.68
	*R* ^2^	2.9%			
Step II	Emotional abuse	16.16	2.59	0.42	6.24 ***
	Δ*R*^2^	17.6% **			
Step III	Emotional abuse	9.89	2.43	0.26	4.06 ***
	Avoidant attachment	9.57	1.35	0.44	7.08 ***
	Anxious attachment	2.41	1.29	0.11	1.87 *
	Δ*R*^2^	19.4%			
Step IV	Emotional abuse	6.21	2.55	0.16	2.44 **
	Avoidant attachment	8.28	1.36	0.38	6.02 ***
	Anxious attachment	1.34	1.29	0.06	1.03
	Rejection expectancy	2.06	1.53	0.89	1.34
	Acceptance expectancy	−6.32	1.94	−0.23	−3.265 ***
	*R* ^2^	4.7% *			

Note: * *p* < 0.05, ** *p* < 0.01, *** *p* < 0.001. Men—1 women—0.

**Table 3 ijerph-21-01679-t003:** Mediating results between the independent variables and fear of intimacy.

Independent	Mediator	Dependent	Independent → Mediator	Mediator → Dependent	Independent → Dependent ^1^	Direct Effect	Indirect Effect	95% Confidence Interval
Childhood emotional abuse	Anxious attachment	Rejection expectancy	0.30	0.32 ***	0.33	0.14	0.10	0.005, 0.20
Childhood emotional abuse	Avoidant attachment	Acceptance expectancy	0.73 ***	−0.27 ***	−0.51 **	−0.25	−0.20	−0.34, −0.07
Childhood emotional abuse	Anxious attachment	Acceptance expectancy	0.30	−0.19 **	−0.51 **	−0.25	−0.06	−0.14, −0.00
Childhood emotional abuse	Avoidant attachment	Fear of intimacy	0.73 ***	0.43 ***	0.72 ***	0.34 *	0.32	0.15, 0.50
Childhood emotional abuse	Anxious attachment	Fear of intimacy	0.30	0.17 **	0.72 ***	0.34 *	0.05	0.00, 0.13
Avoidant attachment	Acceptance expectancy	Fear of intimacy	−0.30 ***	−0.26 ***	0.48 ***	0.40 ***	0.08	0.03, 0.15
Anxious attachment	Acceptance expectancy	Fear of intimacy	−0.20 **	−0.26 ***	0.18 **	0.13 *	0.05	0.01, 0.11

* *p* < 0.05, ** *p* < 0.01, *** *p* < 0.001, ^1^ Independent, → Dependent, Total effect = Direct + Indirect Effects.

## Data Availability

The data presented in this study are available upon request from the corresponding authors.

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
