# Peer review of "From Emotional Abuse to a Fear of Intimacy: A Preliminary Study of the Mediating Role of Attachment Styles and Rejection Sensitivity"

_ijerph, 2024, doi:10.3390/ijerph21121679_

Round 1

Reviewer 1 Report

Comments and Suggestions for Authors

The paper proposes to examine the mediation role of childhood emotional abuse and reaction sensitivity between insecure attachment and fear of intimacy. The authors proposed a hierarchical regression and a process function to probe such relationships. They conclude that avoidant and anxious attachment mediated the relationship between childhood emotional abuse and fear of intimacy. they recommended creating interventions to promote earning security.

Introduction

Authors might resume and establish what is known about the actual relationship between the suggested variables and make clear what innovative statistical methods have been used to examine the proposed relationship between referred variables.

Consistency about the objective referred to in the abstract and the introduction might be solved.

Method

The method section might refer to all questionaries' previous validity data, not only Cronbach alphas.

Authors might refer to how they used the criteria for internet E-surveys, such as data protection, development, testing, contact mode, advertising the survey, compulsory/voluntary participation, completion rate, cookies used, IP check, log file analysis, registration, and atypical timestamp considerations. I suggest reviewing the recommendations of Eysenbach (2004). I also recommend describing how authors eliminated respondents who failed to complete the survey since the survey platform automatically eliminated them.

The main recommendation is the data analysis. According to the study's objective, advanced statistical procedures might be able to reach valid conclusions about the relationship between variables and according to the hypothesis. I suggest using confirmatory factor analysis to evaluate the structure of the latent proposed variables, a structural equational model to assess relationships planted through hierarchical regression analysis, and measurement invariances, if possible, to probe if there are no biases in measures of the variables.

Results

The results might be valid if the statistical analysis is corrected. Additionally, results are related only to correlations, low betas, and unclear procedures for constructing the model in Figure 1.

Discussion

Therefore, affirmation about mechanisms linking childhood emotional abuse and later outcomes in adulthood in terms of forming intimate relationships is unclear.

The transactional—non-experimental study does not allow the authors to propose the assumptions made in the discussion. Even using an advanced statistical model, authors might only propose associations but no causal relationship between the study's variables.

References

Only 26% of the references have less than five years. Authors might consider recent evidence about the variables suggested in the whole paper.

Comments on the Quality of English Language

I suggest reviewing the grammar in the whole paper at the end of correction attendance.

Author Response

Introduction

Authors might resume and establish what is known about the actual relationship between the suggested variables and make clear what innovative statistical methods have been used to examine the proposed relationship between referred variables.

Thank you for your suggestion. The first paragraph of the introduction was changed accordingly. In addition, reference to the statistical methods employed for examining the hypotheses is presented wherever applicable.

Consistency about the objective referred to in the abstract and the introduction might be solved.

Thank you for this remark, the inconsistencies have been corrected.

The method section might refer to all questionaries' previous validity data, not only Cronbach alphas.

Thank you for this remark, this has been corrected.

Authors might refer to how they used the criteria for internet E-surveys, such as data protection, development, testing, contact mode, advertising the survey, compulsory/voluntary participation, completion rate….recommendations of Eysenbach (2004).

Thank you for this remark, the study does indeed comply with Eysenbach’s (2004) guidelines: Informed Consent, Confidentiality, Anonymity, Data Security, Ethical Considerations, as mention in the text.

Moreover, according to Qualtrics’ website, the software employs several robust measures to handle sensitive data securely:

  1. Encryption: All data in transit is encrypted using HTTPS and HTTP Strict Transport Security (HSTS) to protect against eavesdropping and session hijacking1.
  2. Data Redaction and Restriction: Qualtrics allows for the redaction and restriction of sensitive data or Personally Identifiable Information (PII) across the organization1.
  3. Compliance with Standards: Qualtrics is compliant with various industry standards and certifications, including ISO 27001, FedRAMP, HITRUST, and SOC2 Type 21. These certifications ensure best practices in information security, risk management, and data protection.

I suggest using confirmatory factor analysis to evaluate the structure of the latent proposed variables, a structural equational model to assess relationships planted through hierarchical regression analysis, and measurement invariances, if possible, to probe if there are no biases in measures of the variables.

Thank you for this remark, we concur that the application of structural equation modeling presents benefits in many complex models. Nevertheless, given that our model does not postulate specific latent factors within the dataset, and the estimated model is saturated (with zero degrees of freedom), fit statistics are unlikely to provide a meaningful contribution to our conclusions, considering the disparities between employing the SEM framework and the Process macro (Rockwood & Hayes, 2020).

We therefore chose to keep the analyses performed.

The results might be valid if the statistical analysis is corrected. Additionally, results are related only to correlations, low betas, and unclear procedures for constructing the model in Figure 1.

Thank you for your comment; however, for the reasons stated in the previous response, we chose to keep the analyses performed intact.

The authors might only propose associations but no causal relationship between the study's variables.

Thank you for this remark, we have corrected all instances in which causality was implied. Furthermore, we referred to the cross-sectional nature of this study in the Limitations section.

Only 26% of the references have less than five years. Authors might consider recent evidence about the variables suggested in the whole paper.

Thank you for this suggestion. Indeed, there are some updated studies (e.g. Vaillancourt-Morel et al’s (2024) meta-analysis); however, very few focus on specifically examining the association between the variables studied, which is why we were unable to include many that are less than five years old.

Reviewer 2 Report

Comments and Suggestions for Authors

Thank you for the opportunity to review the article entitled "From emotional abuse to fear of intimacy: The mediating role of attachment styles and rejection sensitivity." The article assesses mechanisms by which emotional abuse in childhood leads to fears of intimacy in adulthood by assessing the indirect roles of insecure attachment and rejection sensitivity. While the article is well-written and highlights the long-lasting effects of childhood emotional abuse, I have concerns about the sampling/recruitment strategy with regards to data validity and bias.

Introduction:

-First paragraph should include a definition of intimacy

-I find the second sentence to be unnecessary and painting too broad a stroke

-Rejection sensitivity should be explicitly defined, including descriptions of acceptance expectancy and rejection expectancy.

Methods:

- I find the sampling method of relying solely on the researchers’ social networks to recruit participants to be concerning, especially given the sensitive nature of the topics studied (i.e. childhood abuse). This introduces significant bias into the sample and raises significant questions about the validity of the data, even with anonymized responses.

Results:

-In step one of the model, the authors say the analysis indicated, “albeit insignificantly”, that women had greater fear of intimacy. If the analysis was not significant, would the analysis not indicate that women and men do not differ in fear of intimacy?

-To aid with clarity, authors should include directionality when describing the results of the mediation analyses. For example, “the higher the emotional abuse is, the higher the avoidant or anxious attachment, which affects fear of intimacy.”

Discussion:

-Again, for clarity purposes, in the first paragraph of the discussion, directionality of the results should be described.

Author Response

Introduction:

-First paragraph should include a definition of intimacy

Thank you for this useful remark, a definition of intimacy has been added.

-I find the second sentence to be unnecessary and painting too broad a stroke

Thank you for this suggestion, this sentence has been deleted.

-Rejection sensitivity should be explicitly defined, including descriptions of acceptance expectancy and rejection expectancy.

We appreciate this suggestion, a definition and some clarification of the construct have been added accordingly.

Methods:

- I find the sampling method of relying solely on the researchers’ social networks to recruit participants to be concerning, especially given the sensitive nature of the topics studied (i.e. childhood abuse). This introduces significant bias into the sample and raises significant questions about the validity of the data, even with anonymized responses.

Thank you for this comment. We have made the corrections suggested and stated as much in the Limitations section.

Nevertheless, with 29% of our study participants reporting having experienced emotional abuse, our study does seem to align with a similar one conducted in Israel, as well as with Stoltenborgh and colleagues’ (2012) meta-analysis of the prevalence of emotional abuse. This reflects well on our data validity, at least in this respect.

Results:

-In step one of the model, the authors say the analysis indicated, “albeit insignificantly”, that women had greater fear of intimacy. If the analysis was not significant, would the analysis not indicate that women and men do not differ in fear of intimacy?

Thank you for this remark, this sentence has been corrected.

-To aid with clarity, authors should include directionality when describing the results of the mediation analyses. For example, “the higher the emotional abuse is, the higher the avoidant or anxious attachment, which affects fear of intimacy.”

Thank you for this suggestion, it has been incorporated into the manuscript.

Discussion:

-Again, for clarity purposes, in the first paragraph of the discussion, directionality of the results should be described.

Thank you for this comment, this suggestion has been incorporated into the manuscript.

Reviewer 3 Report

Comments and Suggestions for Authors

Finzi-Dottan and Abadi present a well designed work on a very interesting issue, that of the mediating role 2 of attachment styles and rejection sensitivity.

The design is appropriate. 

The questionnaires had been validated to the targeted population? If not, this is a limitation of the study that should be mentioned in the limitations section.

The statistics although good, lack description and references. In section 2.3 they have to explain comprehensively their tools, how they report the results, which statistical program they used etc.

Comments on the Quality of English Language

No significant plagiarism has been detected. the language is generally good; minor editing needed in the discussion section. In detail, they should avoid large and complex sentences.

Author Response

The questionnaires had been validated to the targeted population?

Thank you for this remark, the questionnaires have indeed been validated to the targeted population and this information was added in context.

The statistics although good, lack description and references.

Thank you for this remark, information on the statistics has been added.

In section 2.3 they have to explain comprehensively their tools, how they report the results, which statistical program they used etc.

Thank you for this suggestion, a description of the study tools has been added, and one of the statistical software used now appears in the Data Analysis section.

Comments on the Quality of English Language

needed in the discussion section. In detail, they should avoid large and complex sentences.

Thank you for this comment, the Discussion section has been re-edited.

Round 2

Reviewer 1 Report

Comments and Suggestions for Authors

I consider that the manuscript has been sufficiently improved to warrant publication in IJERPH. Thank you for taking my comments into account.

Author Response

We thank you for your review

Reviewer 2 Report

Comments and Suggestions for Authors

Thank you again for the opportunity to review the manuscript entitled "From emotional abuse to fear of intimacy: The mediating role of attachment styles and rejection sensitivity". While I appreciate the author's responsiveness, I still have concerns about the significant bias introduced into the data by relying on personal social networks for data collection, and do not think this point is sufficiently addressed. 

Author Response

We thank you very much, we will correct this important point in accordance with the editor's instructions. 
